# Parameter Inference
# with Bifurcation Diagrams

**Gregory Szep**
King's College London
London, WC2R 2LS, UK
`gregory.szep@kcl.ac.uk`

**Attila Csikász-Nagy**
Pázmány Péter Catholic University
Budapest, 1083, Hungary
`csikasznagy@gmail.com`

**Neil Dalchau**
Microsoft Research Cambridge
Cambridge, CB1 2FB, UK
`ndalchau@gmail.com`

## Abstract

Estimation of parameters in differential equation models can be achieved by applying learning algorithms to quantitative time-series data. However, sometimes it is only possible to measure qualitative changes of a system in response to a controlled condition. In dynamical systems theory, such change points are known as *bifurcations* and lie on a function of the controlled condition called the *bifurcation diagram*. In this work, we propose a gradient-based approach for inferring the parameters of differential equations that produce a user-specified bifurcation diagram. The cost function contains an error term that is minimal when the model bifurcations match the specified targets and a bifurcation measure which has gradients that push optimisers towards bifurcating parameter regimes. The gradients can be computed without the need to differentiate through the operations of the solver that was used to compute the diagram. We demonstrate parameter inference with minimal models which explore the space of saddle-node and pitchfork diagrams and the genetic toggle switch from synthetic biology. Furthermore, the cost landscape allows us to organise models in terms of topological and geometric equivalence.

## 1   Introduction

Inverse problems [1] arise in biology and engineering in settings when the model is not fully known and the desire is to match model behaviour to a given set of observations. This helps systematically guide both model and experimental design. While we would like to understand the quantitative details of a system, often only qualitative changes in response to varying experimental conditions can be robustly measured across independent studies [2, 3]. For example, several studies are likely to agree that the human immune system activates above a threshold concentration of a pathogen and deactivates at a lower threshold concentration, but may disagree on the exact quantities of the thresholds or the magnitudes of the immune response. Bifurcation theory provides us a framework for studying these transitions in a manner that is independent of quantitative details [4]. The emerging picture suggests that identification of the qualitative behaviour – the bifurcation diagram – should precede any attempt at inferring other properties of a system [5].

Inferring the parameters of a model directly from a bifurcation diagram is difficult because it is not obvious how multiple parameters in concert control the existence and position of a bifurcation. It could even be impossible for the model to bifurcate in the manner desired. For models with a sufficiently small number of parameters, finding specific bifurcation diagrams is typically done by

hand [6]. Several approaches exist to place bifurcations to desired locations once a manifold is present [7–9] yet typically resort to sampling techniques to search for them in the first place [10, 11]. It is always possible to design bespoke goodness of fit measures to find specific model behaviours, for example using the period and phase of limit cycle oscillations [12]. However, this approach does not generalise across a wider set of qualitative behaviours. Progress has been made in cases where model structure and stability conditions are used to refine the search space [13, 14] yet the resulting objectives are still not explicit in the bifurcation targets and also not differentiable. In the emerging field of scientific machine learning [15–17], parameters of structured mechanistic models are favoured over flexible models in larger parameter spaces. A scalable method for navigating the space of bifurcation diagrams would enable design of differential equations with high-level qualitative constraints. Furthermore one could begin organising models according to qualitatively distinct behaviours.

Back-propagation through differential equation solvers has been a breakthrough over the past couple of years [18, 19] that enabled scalable parameter inference for differential equations from trajectory data. Although one could use trajectory data to create the aforementioned qualitative constraints [20, 21] this would entail over-constraining information originating from the kinetics and dynamical transients of the model. Furthermore, such data usually does not contain sufficient information about dynamical transients in order to identify kinetic parameters. Techniques for back-propagating through implicit equation solvers have also been developed [22, 23] although to the best of the authors' knowledge have not been applied to bifurcation diagrams at the time of writing this paper.

The problem of inferring differential equation parameters against a user-specified bifurcation diagram decomposes into two parts: searching for bifurcating regimes and matching the locations of bifurcation points to desired values. Matching bifurcation locations is a supervised problem where the data are expressed as bifurcations points [8, 11]. Searching for bifurcations is an unsupervised problem because when bifurcations are not present, there is no distance defined between data and prediction [10]. Therefore only properties of the model can be used to start the search. We propose an approach for performing both tasks in an end-to-end fashion. The bifurcation diagram encodes high-level qualitative information defined by state space structures, rather than kinetics. We apply the strategy of implicit layers [22, 23] to calculate gradients. To compute the diagram we use a predictor-corrector method called deflated continuation [24, 25] developed for partial differential equations.

We find that the cost function landscape contains basins that not only allow us to synthesise models with a desired bifurcation diagram but also allow us to organise models in terms of topological and geometric equivalence. We discuss the relevance of this in model selection. In summary, our paper has the following main contributions:

- An end-to-end differentiable method for locating bifurcations in parameter space and then matching their dependency on a control condition to user-specified locations
- Implementation of the method as a Julia package `BifurcationInference.jl`
- Leveraging the cost landscape for a novel way of organising differential equation models in terms of geometric and topological equivalence

## 1.1 Preliminaries

Suppose we collected observations along a scalar control condition $p \in \mathbb{R}$ and conclude that there are specific values of $p$ for which there are qualitative changes in system behaviour. Let $\mathcal{D}$ be the set of those values and let us hypothesise that these transitions occur due to bifurcations in the dynamics that drive the underlying mechanism. Let us model the mechanism with a parametrised set of differential equations for states $u \in \mathbb{R}^N$ with a vector function $F_\theta$ in a parameter space $\theta \in \mathbb{R}^M$.

For the purposes of introducing this work, we will consider the simplest class of bifurcations known as *co-dimension one* bifurcations not including limit cycles. Therefore $\mathcal{D}$ should contain conditions for which we hypothesise changes in multi-stable behaviour. Let the equations be

$$\frac{\partial u}{\partial t} = F_\theta(u, p) \qquad \text{where} \quad F_\theta : \mathbb{R}^{N+1} \to \mathbb{R}^N \tag{1}$$

In the context of the differential equations, and not considering limit cycles for now, we show that a static non-degenerate bifurcation can be defined by a set of conditions on the determinant of the Jacobian $\left| \frac{\partial F_\theta}{\partial u} \right|$. The determinant of the Jacobian quantifies the rate at which trajectories in a local

patch of state-space $u \in \mathbb{R}^N$ converge or diverge. Let $s \in \mathbb{R}$ parametrise the curves that trace out the bifurcation diagram. Any location on the curve $u(s)$ and $p(s)$ must satisfy the steady-state of equations (1). Directional derivatives $\frac{d}{ds}$ along the diagram require the calculation of a vector that is tangent to the diagram (see Supplementary A). The determinant approaching zero along the diagram means that the dynamics of the system are slowing down, which is an important indicator for the onset of a transition between qualitative behaviours. Furthermore, the slowing down must necessarily be followed by a breakdown of stability; for this to be true it is sufficient *but not necessary* to require that the determinant cross zero with a finite slope, meaning that its directional derivative along the diagram $\frac{d}{ds}\left|\frac{\partial F_\theta}{\partial u}\right|$ is not zero. This is the non-degeneracy condition. The set of predicted values for the control condition $\mathcal{P}(\theta) \subset \mathbb{R}$ at which bifurcations occur are defined as

$$\mathcal{P}(\theta) := \left\{ p \mid \exists\, u: \ F_\theta(u, p) = 0, \ \left|\frac{\partial F_\theta}{\partial u}\right| = 0, \ \frac{d}{ds}\left|\frac{\partial F_\theta}{\partial u}\right| \neq 0 \right\} \tag{2}$$

A proof of how the conditions (2) are necessary and sufficient for static non-degenerate bifurcations is detailed in Supplementary B. The most common bifurcations between steady states, not including limit cycles, are saddle-nodes and pitchforks [26]. Saddle-node bifurcations, which often appear in pairs (Figure 1A) are defined by stable and unstable fixed points meeting and disappearing. Pitchfork bifurcations occur where a single steady state splits into two stable and one unstable steady state (Figure 1B shows an *imperfect* pitchfork; a *perfect* pitchfork arises when $\theta_1 = 0$). To illustrate these bifurcations, we define minimal models (Figure 1) that span the space of saddle-node and pitchforks, where indeed zero crossings in the determinant with a finite slope define the set of prediction $\mathcal{P}(\theta)$. The location of these crossings in general may not match the targets $\mathcal{D}$.

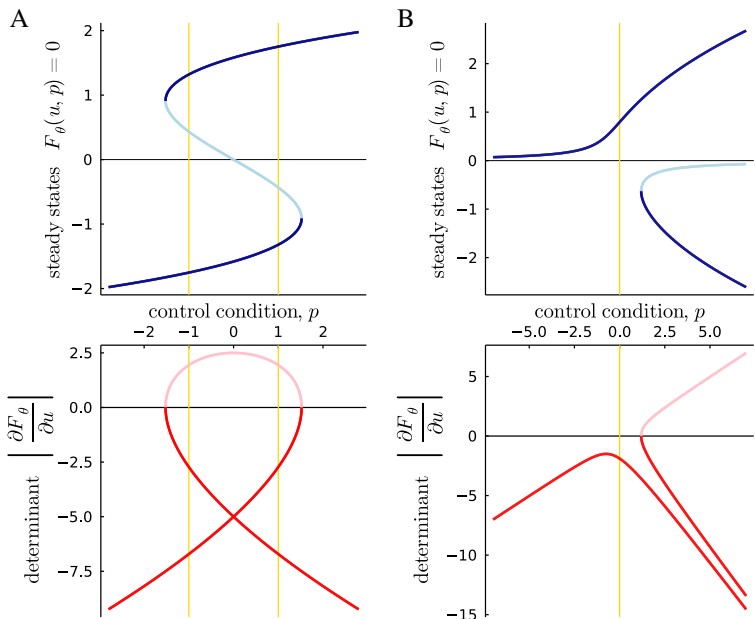

Figure 1: Illustration of bifurcation diagrams for minimal models of bifurcations. A. Saddle-node bifurcations arise for $F_\theta(u, p) = p + \theta_1 u + \theta_2 u^3$ when $\theta = (\frac{5}{2}, -1)$. B. Pitchfork bifurcations arise for $F_\theta(u, p) = \theta_1 + pu + \theta_2 u^3$ when $\theta = (\frac{1}{2}, -1)$. Targets are illustrated by light yellow vertical lines. Bifurcation curves are shown as solid blue and red lines, with lighter shades indicating the determinant crossing zero at locations $\mathcal{P}(\theta)$ giving rise to unstable solutions.

For a given set of parameters $\theta$ one could compute the set of predicted bifurcations $\mathcal{P}(\theta)$ using parameter continuation methods [25, 24]. Our goal is to find optimal parameters $\theta^*$ that match predictions $\mathcal{P}(\theta^*)$ to specified targets $\mathcal{D}$. We must design a suitable cost function $L$ so that

$$\theta^* := \operatorname{argmin}_\theta L(\theta|\mathcal{D}) \tag{3}$$

The optimal $\theta^*$ is not expected to always be unique, but is in general a manifold representing the space of qualitatively equivalent models. Ideally, the cost function $L$ should reward $\theta$ for which the number of predicted bifurcations is equal to the number of targets, $|\mathcal{P}(\theta)| = |\mathcal{D}|$. This is especially important in the case where there are no predictions $|\mathcal{P}(\theta)| = 0$.

## 2 Proposed Method

### 2.1 Cost Function

To identify parameter sets that give rise to bifurcation diagrams with specified bifurcation points, we propose a cost function that comprises two terms. The role of the error term is simply to reward predicted bifurcations to coincide with the specified target locations. This of course relies on such bifurcations existing. The role of the eigenvalue term is to encourage an optimiser to move towards parameter regimes that do exhibit bifurcations.

#### 2.1.1 Error term: matching bifurcations to target locations

In order for predicted bifurcations $p(\theta) \in \mathcal{P}(\theta)$ to match targets $p' \in \mathcal{D}$ we need to evaluate an error term $|p(\theta) - p'|$. A naive approach might take an average over the norms for all prediction-target pairs. However this gives rise to unwanted cross-terms and the possibility of multiple predictions matching the same target without any penalty for unmatched targets. Therefore, we choose a geometric mean over the predictions and an arithmetic mean over targets:

$$E(\theta, \mathcal{D}) = \frac{1}{|\mathcal{D}|} \sum_{p' \in \mathcal{D}} \prod_{p(\theta) \in \mathcal{P}(\theta)} |p(\theta) - p'|^{\frac{1}{|\mathcal{P}|}} \tag{4}$$

The error term is only zero when each target is matched by at least one prediction and allows for cases where the number of predictions is greater than or equal to the number of targets $|\mathcal{P}| \geq |\mathcal{D}|$. An alternative approach, which undesirably introduces more hyper-parameters, would be to let each prediction $\mathcal{P}(\theta)$ represent the centroid of a mixture distribution and use expectation-maximisation to match the centroids to targets $\mathcal{D}$.

#### 2.1.2 Eigenvalue term: encouraging bifurcations

We can see from Figure 1 and definition (2) that predictions $p(\theta)$ can be identified by looking for points along the curve where the determinant crosses zero $\left|\frac{\partial F_\theta}{\partial u}\right| = 0$ with a finite slope $\frac{d}{ds}\left|\frac{\partial F_\theta}{\partial u}\right| \neq 0$. Using these quantities we can define a positive semi-definite measure $\varphi_\theta(s)$ of zero crossings in the determinant along a curve parametrised by $s$ which we define as

$$\varphi_\theta(s) := \left(1 + \left|\frac{\left|\frac{\partial F_\theta}{\partial u}\right|}{\frac{d}{ds}\left|\frac{\partial F_\theta}{\partial u}\right|}\right|\right)^{-1} \tag{5}$$

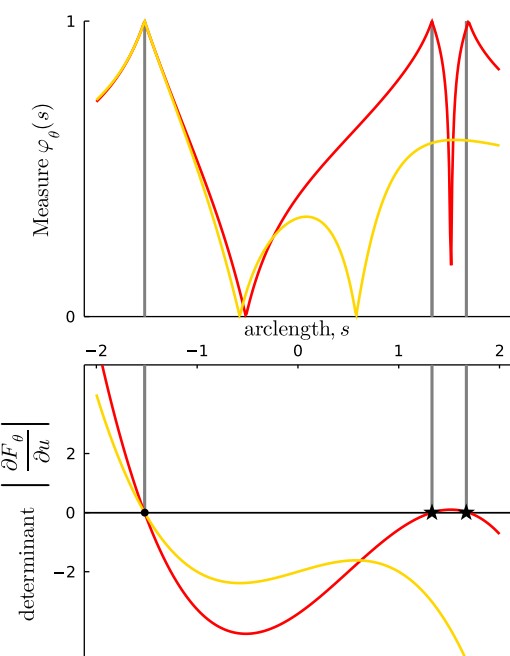

The bifurcation measure $\varphi_\theta(s)$ is maximal at bifurcations and has finite gradients in non-bifurcating regimes (Figure 2). More specifically, the measure $\varphi_\theta(s)$ is one at bifurcation points and goes to zero an odd number of times between bifurcations. This is because $\left|\frac{\partial F_\theta}{\partial u}\right|$ must eventually turn around in order to return back to zero, resulting in the directional derivative $\frac{d}{ds}\left|\frac{\partial F_\theta}{\partial u}\right|$ going to zero. Hence the measure $\varphi_\theta(s)$ goes to zero for each turning point (see Figure 2).

On the other hand, as the determinant $\left|\frac{\partial F_\theta}{\partial u}\right|$ diverges, we approach regimes far away from any bifurcations and hence $\varphi_\theta(s) \to 0$. Since we would still like to have non-zero gradients with respect to $\theta$ in these regimes we designed the measure to go to zero sufficiently slowly.

Figure 2: Bifurcation measure $\varphi_\theta(s)$ and determinant $\left|\frac{\partial F_\theta}{\partial u}\right|$ along the arclength $s$ of two different bifurcation curves demonstrating how maximising the measure along the curve maintains the existing bifurcation marked by a circle, while encouraging new bifurcations marked by stars.

While the calculation of the determinant is straightforward, its directional derivative requires a tangent vector to the bifurcation curve. Fortunately the tangent vector $T_\theta(s)$ at the solution $u(s), p(s)$ anywhere along the curve $s$ can be calculated as the nullspace of the rectangular $N \times (N+1)$ Jacobian

$$\left. \frac{\partial F_\theta}{\partial(u,p)} \right|_{F_\theta(u(s),p(s))=0} \cdot T_\theta(s) = 0 \tag{6}$$

This equation guarantees that the tangent vector $T_\theta(s)$ is orthogonal to all hyper-planes defined by the components of $F_\theta$. In this setting the dimension of the nullspace is always known, and therefore can reliably be calculated using QR factorisation methods [27].

Equipped with a measure that quantifies the appearance of bifurcations along a bifurcation arc we can define the total measure for a bifurcation diagram as

$$\Psi(\theta) := \frac{\int_{F_\theta(u,p)=0} \varphi_\theta(s)\,\mathrm{d}s}{\int_{F_\theta(u,p)=0} \mathrm{d}s}. \tag{7}$$

Here we denote $\int_{F_\theta(u,p)=0} \mathrm{d}s$ as the sum of the line integrals in $(u,p) \in \mathbb{R}^{N+1}$ defined by the level set $F_\theta(u,p) = 0$ with $s$ being an arbitrary parametrisation of the curves. The total measure $\Psi(\theta)$ is normalised such that $\Psi(\theta) \to 1$ in the regimes where the controlled condition region $p$ is densely packed with bifurcations. The total measure $\Psi(\theta)$ is added to the error term as if it were a likelihood. This defines the cost function as

$$L(\theta|\mathcal{D}) := \big(|\mathcal{P}| - |\mathcal{D}|\big) \log \Psi(\theta) + E(\theta, \mathcal{D}), \tag{8}$$

The pre-factor $|\mathcal{D}| - |\mathcal{P}|$ in the eigenvalue term ensures that the gradients are always pushing optimisers towards a state where $|\mathcal{D}| = |\mathcal{P}|$. This can be seen as a step-wise annealing of the eigenvalue term until the desired state is reached.

## 2.2 Differentiating the cost function

To make use of gradient-based optimisers to locate desired bifurcation diagrams, we show here how to differentiate the cost function. First, we note that while individual bifurcations $p(\theta)$ depend smoothly on $\theta$, the total number of predictions $|\mathcal{P}|$ does not have gradient contributions with respect to $\theta$. Therefore, we can safely drop the dependency in the prediction counter and now proceed in taking gradients with respect to $\theta$ knowing that the only dependencies we need to track are for individual bifurcations $p(\theta)$ within the definition the error term (4) and the total measure (7). Therefore,

$$\frac{\partial L}{\partial \theta} = \big(|\mathcal{P}| - |\mathcal{D}|\big) \lambda \frac{\partial \Psi}{\partial \theta} \Psi(\theta)^{-1} + \frac{1}{|\mathcal{D}||\mathcal{P}|} \sum_{p'} \prod_{p(\theta)} |p(\theta) - p'|^{\frac{1}{|\mathcal{P}|}} \sum_{p(\theta)} \frac{\partial p}{\partial \theta} (p(\theta) - p')^{-1} \tag{9}$$

In a similar vein to back-propagation through neural differential equations [18] we would like to be able to calculate the gradient $\frac{\partial L}{\partial \theta}$ without having to differentiate through the operations of the solver that finds the bifurcation diagram $F_\theta(u,p) = 0$ and the bifurcation locations $p(\theta)$. To calculate the gradient of the measure $\frac{\partial \Psi}{\partial \theta}$ we need to differentiate line integrals that depend on $\theta$. Fortunately this can be done by the application of the generalised Leibniz integral rule, details of which can be found in Supplementary C.

The gradient of the bifurcation points $\frac{\partial p}{\partial \theta}$ is found by application of the implicit function theorem to a vector function $G_\theta : \mathbb{R}^{N+1} \to \mathbb{R}^{N+1}$ whose components represent the two constraints $F_\theta(u,p) = 0$ and $\left| \frac{\partial F_\theta}{\partial u} \right| = 0$. By following a similar strategy to that used by implicit layers [22] we yield an $(N+1) \times M$ Jacobian representing a deformation field [28] for each $\theta$ direction. The gradient we are looking for becomes

$$\frac{\partial p}{\partial \theta} = -\hat{p} \cdot \left. \frac{\partial G_\theta}{\partial(u,p)}^{-1} \frac{\partial G_\theta}{\partial \theta} \right|_{G_\theta(u,p)=0} \quad \text{where} \quad G_\theta(u,p) := \begin{bmatrix} F_\theta(u,p) \\ \left| \frac{\partial F_\theta}{\partial u} \right| \end{bmatrix} \tag{10}$$

Here $\hat{p}$ is a unit vector in $(u,p) \in \mathbb{R}^{N+1}$ that picks out the deformations along the $p$-direction. If we wanted to place the bifurcation at target steady state $u'$ as well as target control condition $p'$ we would use the full $(N+1) \times M$ deformation matrix. Calculation of this matrix involves inverting an

$(N+1) \times (N+1)$ Jacobian $\frac{\partial G_\theta}{\partial(u,p)}$. Instead of explicitly inverting the Jacobian the corresponding system of linear equations is solved. The determinant of this Jacobian goes to zero in the degenerate case where $\frac{d}{ds}\left|\frac{\partial F_\theta}{\partial u}\right| = 0$, further justifying our choice of measure $\Psi(\theta)$ which discourages the degenerate case.

The cost function is piece-wise smooth and differentiable with undefined gradients only in parameter contours where the number of predictions $|\mathcal{P}|$ changes; this is when $\Psi(\theta)$ is undefined and the inverse of $\frac{\partial G_\theta}{\partial(u,p)}$ does not exist. Given a set of solutions to $F_\theta(u,p) = 0$ and locations $p(\theta)$ the gradient $\frac{\partial L}{\partial \theta}$ can be evaluated using automatic differentiation methods [29–31] without needing to back-propagate through the solver that obtained the level set $F_\theta(u,p) = 0$ in the forward pass.

# 3  Experiments & Results

In this section, we apply the method first to minimal examples that can produce saddle-node and pitchfork bifurcations (both $N = 1$, $M = 2$), and then a slightly more complex model ($N = 2$, $M = 5$) that has multiple parametric regimes producing saddle-node bifurcations. We also demonstrate our method on a model of greater complexity, to convince the reader that the method can be used on more realistic examples with practical significance. In Supplementary D we demonstrate the identification of saddle-node bifurcations and damped oscillations in a model ($N = 4$, $M = 21$) of a synthetic gene circuit in *E. coli* [3].

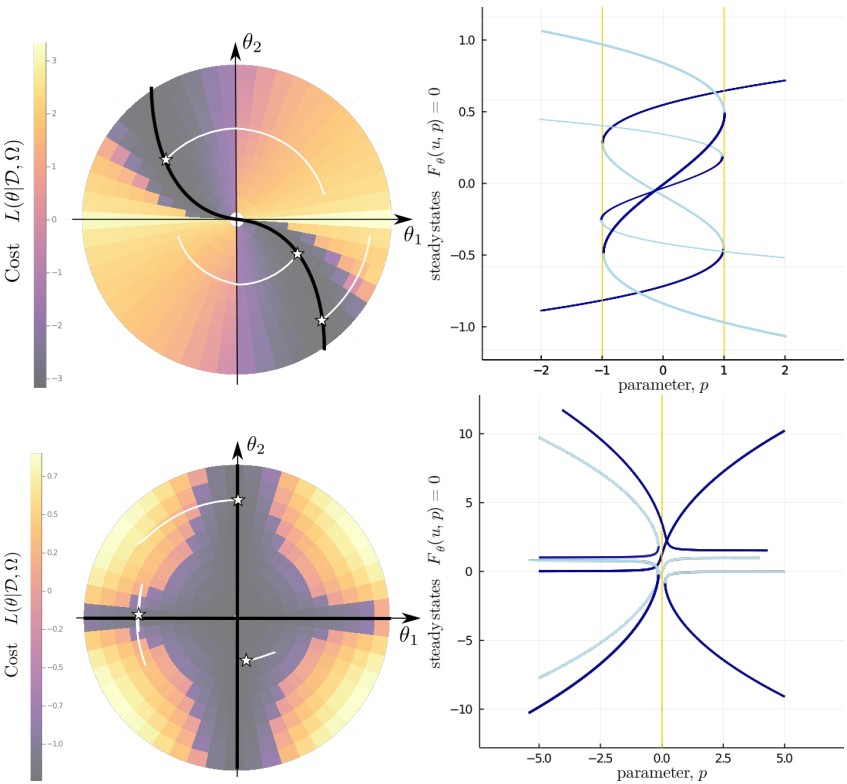

Figure 3: Saddle-node $F_\theta(u,p) = p + \theta_1 u + \theta_2 u^3$ and pitchfork $F_\theta(u,p) = \theta_1 + up + \theta_2 u^3$ optimised with respect to $\theta$ so that predicted bifurcations $\mathcal{P}(\theta)$ match targets $\mathcal{D}$ in control condition $p$. The right panel shows bifurcations diagrams for the three optimal $\theta^*$ marked by stars on the left panel. The optimisation trajectories in white follow the gradient of the cost, approaching the black lines of global minima in the left panel

## 3.1  Minimal Models

Optimisations of two parameters $(\theta_1, \theta_2)$ using simple gradient descent from `Flux.jl` with learning rate $\eta = 0.01$ for the minimal saddle-node and pitchfork models (Figure 1) yield trajectories approach-

ing lines of global minima in the cost function (Figures 3) which represent a set of geometrically equivalent models. Two bifurcation diagrams are geometrically equivalent if the number, type and locations of bifurcations match the specified targets $\mathcal{D}$.

We can see that the geometrically equivalent lines are contained within larger basins where the correct number and type of bifurcations are present but do not match the locations of targets $\mathcal{D}$. All models within this basin are in some sense topologically equivalent. This hierarchical classification allows us to identify the set of models that satisfy observed qualitative behaviour [5] before any attempt at inferring kinetic parameters, which is done by choosing a model along the line of geometrically equivalent models.

Optimisation trajectories for the two minimal models appear mostly circumferential. This is because the models were set up such that the radial direction from the origin in $\theta$ space mostly scale kinetics whereas the circumferential direction changes the bifurcation topology. This suggests that the gradients of our cost function seek to change model geometry over kinetics.

## 3.2 Genetic Toggle Switch

In this section we optimise a model where the states share a Hill function relationship with co-operatively $n = 2$; these models often emerge from mass action kinetics with quasi-steady state approximations and are used to model species concentrations. After re-scaling the equations governing the dynamics of concentrations, the simplified equations for state $u_1$ and $u_2$ become

$$\partial_t u_1 = \frac{a_1 + (pu_2)^2}{1 + (pu_2)^2} - \mu_1 u_1 \quad \partial_t u_2 = \frac{a_2 + (ku_1)^2}{1 + (ku_1)^2} - \mu_2 u_2 \tag{11}$$

where $a_k$ is the baseline production rate for species $k$ in the absence of the other species. Each species has a finite degradation rate $\mu_k$. Finally we have two sensitivity constants $p$ and $k$, one of which is chosen as our control condition. A baseline production rate $a_k > 1$ recovers an inhibitor type hill function for species $k$ and is an activator otherwise. The sensitivities are proportional to the slope of the hill productions. Solving for the steady states, substituting the equation for $u_1$ into $u_2$ and rearranging gives rise to the relationship

$$\frac{k}{\mu_1} = \frac{(1 + (\frac{p}{\mu_2}u')^2)\sqrt{a_2 - u'}}{(a_1 + (\frac{p}{\mu_2}u')^2)\sqrt{u' - 1}} \quad \text{where} \quad u' := u_2\mu_2 \tag{12}$$

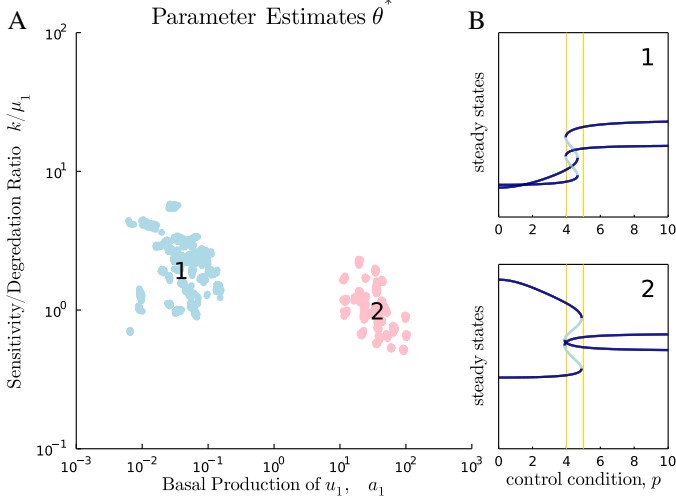

Figure 4: Bifurcation inference for the two-state model (11). A. Optimal parameter estimates $\theta^*$ for the targets $\mathcal{D} = \{4, 5\}$ reveal two clusters of qualitatively different regimes: mutual activation ($a_1 < 1$; cluster 1) and mutual inhibition ($a_1 > 1$; cluster 2). B. Example bifurcation diagrams indicate positively and negatively correlated dependencies between the two model states, as a function of the control condition.

which reveals that only $a_1$, $a_2$ and the ratio between the sensitivity and degradation parameters, $\frac{k}{\mu_1}$, affect the solutions to this equation, and hence the locations of the bifurcations (Figure 4A). In 98% of 800 runs, optimisation using the ADAM optimiser [32] from `Flux.jl` with learning rate $\eta = 0.1$ converged to one of two clusters: mutual activation ($a_1 < 1, a_2 < 1$; cluster 1) and mutual inhibition ($a_1 > 1, a_2 > 1$; cluster 2) regimes. Example bifurcation diagrams illustrate how the bifurcation curves of each species are positively correlated in mutual activation and negatively correlated for mutual inhibition (Figure 4B).

In order to maintain biological interpretability, optimisation was restricted to the positive parameter regime by transforming the parameters to log-space $\theta \to 10^\theta$. At the beginning of each optimisation run an initial $\theta$ was chosen in the log-space by sampling from a multivariate normal distribution with mean zero and standard deviation one.

### 3.3 Complexity

Performing one iteration of the optimisation requires the computation of the gradient of the cost (9), requiring a computation of the bifurcation diagram with parameter continuation methods, which includes the evaluation of matrix inversions (10). Instead of evaluating the inversions directly, we solve a system of linear equations, applying the same strategy as implicit layers [22, 23]. This leaves us with the computational bottleneck of calculating the determinant of the state space Jacobian, required in both the bifurcation measure (5) and gradient (10). This calculation scales like $N^2$ where $N$ is the number of state space variables (Figure 5A).

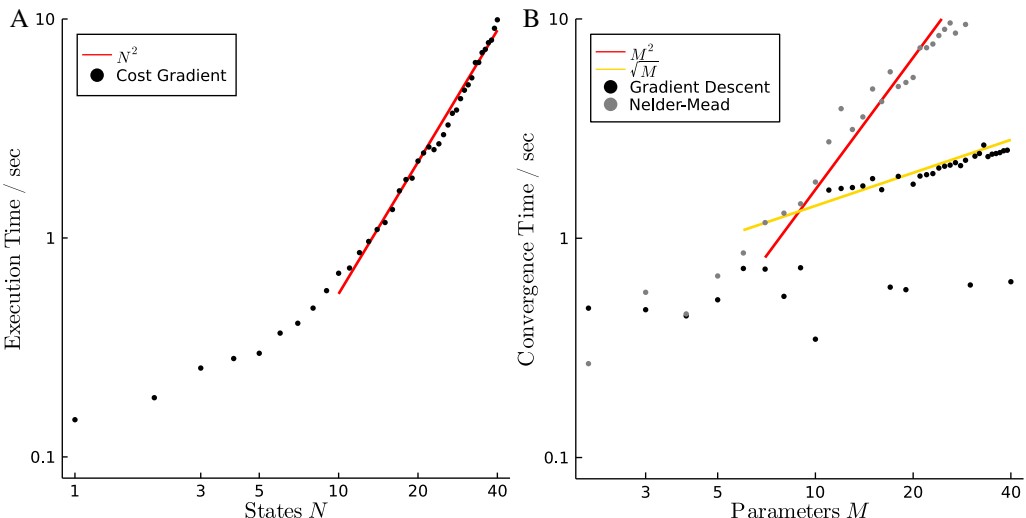

Figure 5: A. Execution time (time to calculate cost gradient) with respect to states $N$. B. Convergence times (the time it takes to find and match a bifurcation to within 1% of a specified target) with respect to the number of parameters $M$, comparing against a gradient-free approach: Nelder-Mead. Calculations were performed on an Intel Core i7-6700HQ CPU @ 2.60GHz x 8 without GPU acceleration.

For the complexity study, a model was designed so that it is extensible both in the number of parameters $M$ and the number of states $N$. There are many choices for this; we opted for a model of the form

$$\begin{cases} \partial_t u_1 = \sin^2 p - (\theta_1 \sin^2 p + 1)u_1 \\ \partial_t u_n = u_{n-1} - (\mu_n^2 + 1)u_n \qquad 2 \le n \le N \end{cases} \tag{13}$$

In this model only the first state $u_1$ defines the shape of the bifurcation diagram, while the remaining states are merely linearly proportional to the first. The parameters $\mu_n$ contain sums of $\theta_m$ allowing us a flexible choice on the number of parameters while maintaining stable solutions for the bifurcation diagram.

While still tractable on laptop computers for states $N < 100$ our implementation currently does not scale well for partial differential equations where a large the number of states $N$ arises from discretisation of the spatial variables. The only reason we need this determinant is because it

is an indicator of bifurcations. We can address the computational bottleneck by finding a more computationally efficient way of calculating this indicator. One approach would be to take the product of a finite subset of eigenvalues of the system. Note that any more efficient calculation must still permit back-propagation through it.

To demonstrate the benefits of the gradient-based aspect of our method we compare convergence times of gradient descent against a gradient-free approach. We use the Nelder-Mead method from `Optim.jl` [33] and obtain convergence times as the number of parameters $M$ is increased (Figure 5B). We observe that for our method convergence times scale like $\sqrt{M}$ compared to $M^2$ for the gradient-free approach.

## 4   Conclusion & Broader Impact

We proposed a gradient-based approach for inferring the parameters of differential equations that produce a user-specified bifurcation diagram. By applying implicit layers [22, 23] and the generalised Leibniz rule [34] to the geometry of the implicitly defined steady states [35] it is possible to use automatic differentiation methods to efficiently calculate gradients. We defined a bifurcation measure that uses the determinant of the state-space Jacobian as an indicator for bifurcating parameter regimes in the eigenvalue term of the cost function. The gradients of the cost can be efficiently computed using automatic differentiation methods. The computational bottleneck is the evaluation of the state-space Jacobian determinant which limits the implementation to ordinary differential equations.

We demonstrated our approach on models with one bifurcation parameter that can give rise to pitchforks and saddle-nodes. The estimated parameters form distinct clusters, allowing us to organise models in terms of topological and geometric equivalence (Figure 3). In the case of the genetic toggle switch (Figure 4) and a more complex model [3] (Figure D.1) we recovered mutual activation and inhibition regimes. In the more complex model we found a damped oscillatory regime that was not known about in the original paper.

Although we did not consider limit cycles, the bifurcation measure can be extended to detect Poincaré-Andronov-Hopf bifurcations alongside changes in stability of fixed points (see Supplementary E for details). This measure enables detection of the onset of damped oscillations and/or the emergence of limit cycles (Figure E.1). Used together with a steady state solver that detects periodic solutions and gradient-based optimisation, we can specify regions of damped oscillation and limit cycles. Our approach generalises naturally to bifurcation manifolds such as limit point curves or surfaces. This is because the normal components of implicit derivatives can still be calculated for under-determined systems of equations [28, 36, 37]. In the case of manifolds it would be more appropriate to use isosurface extraction algorithms rather than continuation to obtain the steady-state manifold. Our approach does not depend on the details of the steady-state solver and therefore can still be applied.

In dynamical systems theory the geometry of state-space determines all of the qualitative behaviours of a system. Our work makes progress towards designing models directly in state-space, rather than the spatial or temporal domain. This is valuable to experimentalists who only have qualitative observations available to them and wish to navigate the space of qualitative behaviours of their system. Our work lies within a trend of progress in the scientific machine learning community, where structured domain-informed models are favoured over flexible models that live in large parameter spaces.

## 5   Acknowledgements

We would like to acknowledge Kieran Cooney for the fruitful conversations that helped guide the derivations and computational approach. A special thanks go to Romain Veltz and the Julia community for helpful pointers on package development and discussions over Slack. This work was supported by Microsoft Research through its PhD Scholarship Programme and the EPSRC Centre for Doctoral Training in Cross-Disciplinary Approaches to Non-Equilibrium Systems (CANES, EP/L015854/1).

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
