# Supplementary Material
# for
# Parameter Inference with Bifurcation Diagrams

**Gregory Szep**
King's College London
London, WC2R 2LS, UK
gregory.szep@kcl.ac.uk

**Attila Csikász-Nagy**
Pázmány Péter Catholic University
Budapest, 1083, Hungary
csikasznagy@gmail.com

**Neil Dalchau**
Microsoft Research Cambridge
Cambridge, CB1 2FB, UK
ndalchau@gmail.com

## Contents

35th Conference on Neural Information Processing Systems (NeurIPS 2021).

# A  Bifurcation Diagrams as Tangent Fields

Let each component of the vector function $F_\theta$ in the model (1) implicitly define a surface embedded in $\mathbb{R}^{N+1}$. Let's assume that the intersection of these $N$ surfaces exists and is not null or degenerate, then the steady states of (1) must be a set of one dimensional space curves in $z \in \mathbb{R}^{N+1}$ defined by

$$F_\theta(z) = 0 \tag{A.1}$$

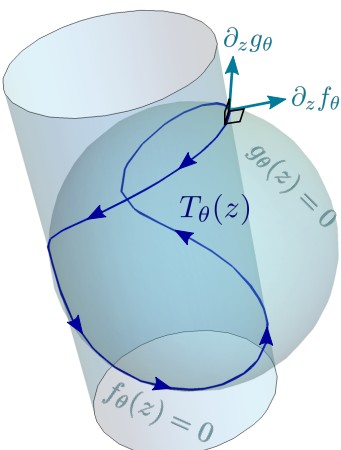

Figure A.1: Two implicit surfaces $f_\theta(z) = 0$ and $g_\theta(z) = 0$ in $\mathbb{R}^3$ intersecting to form a space curve which is tangent to field $T_\theta(z)$ and perpendicular to gradients $\partial_z f_\theta$ and $\partial_z g_\theta$

An expression for the field $T_\theta(z)$ tangent to the set of curves would allow us to take derivatives and integrals along the bifurcation curve. This is exactly what we need to do to evaluate our cost function 8. Fortunately the tangent field can be constructed by ensuring it is perpendicular to the gradient $\partial_z$ of each component of $F_\theta$ as illustrated by an example two component system in Figure A.1. The tangent field $T_\theta(z)$ can be constructed perpendicular to all gradient vectors using the properties of the determinant [35]

$$T_\theta(z) := \begin{vmatrix} \hat{z} \\ \partial_z F_\theta \end{vmatrix} \qquad T_\theta : \mathbb{R}^{N+1} \to \mathbb{R}^{N+1} \tag{A.2}$$

$$= \sum_{i=1}^{N+1} \hat{z}_i (-1)^{i+1} \begin{vmatrix} \dfrac{\partial F_\theta}{\partial(z \setminus z_i)} \end{vmatrix} \tag{A.3}$$

where $\hat{z}$ is a collection of unit basis vectors in the $\mathbb{R}^{N+1}$ space and $\partial_z F_\theta$ is an $N \times (N+1)$ rectangular Jacobian matrix of partial derivatives and $z \setminus z_i$ denotes the $N$ dimensional vector $z$ with component $z_i$ removed. This construction ensures perpendicularity to any gradients of $F_\theta$

$$T_\theta(z) \cdot \partial_z f_\theta = \begin{vmatrix} \partial_z f_\theta \\ \partial_z F_\theta \end{vmatrix} = 0 \quad \forall f_\theta \in F_\theta \tag{A.4}$$

since the determinant of any matrix with two identical rows or columns is zero. Note that the tangent field $T_\theta(z)$ is actually defined for all values of $z$ where adjacent field lines trace out other level sets where $F_\theta(z) \neq 0$. Furthermore deformations with respect to $\theta$ are always orthogonal to the tangent

$$T_\theta(z) \cdot \frac{dT_\theta}{d\theta} = 0 \tag{A.5}$$

Figure A.2 shows how the bifurcation curve defined by $F_\theta(z) = 0$ picks out one of many level sets or traces in tangent field $T_\theta(z)$ for the saddle and pitchfork. The tangent field $T_\theta(z)$ can always be analytically evaluated by taking the determinant in (A.2). We will proceed with calculations on $T_\theta(z)$ in the whole space $z$ and pick out a single trace by solving $F_\theta(z) = 0$ later. For our two models

$$T_\theta(z) = \hat{u} - (3\theta_2 u^2 + \theta_1)\,\hat{p} \qquad\qquad T_\theta(z) = u\hat{u} - (3\theta_2 u^2 + p)\,\hat{p} \tag{A.6}$$
$$\text{saddle−node model} \qquad\qquad\qquad\qquad \text{pitchfork model}$$

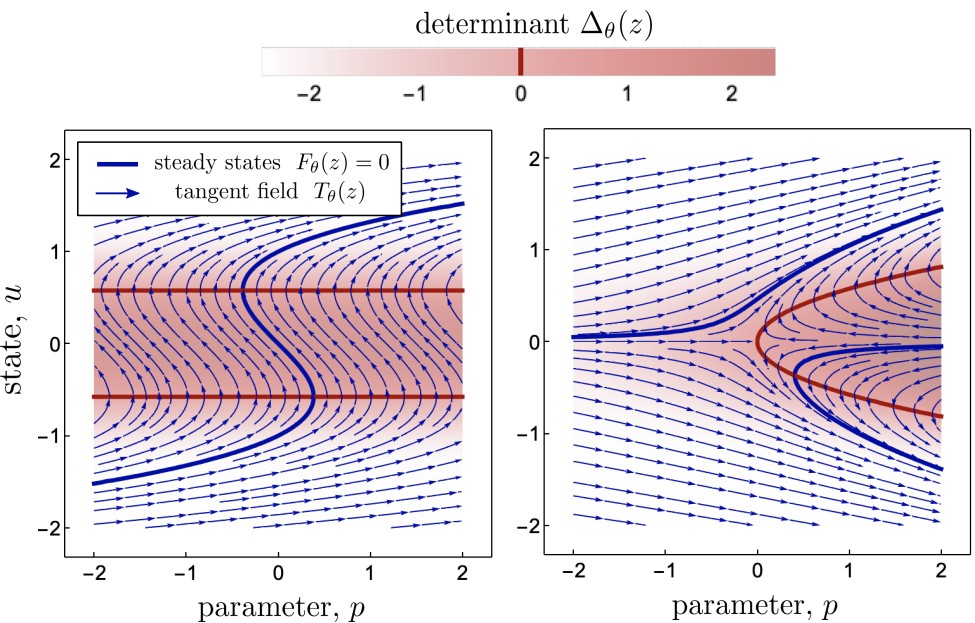

Figure A.2: Left/Right : Determinant $\left|\frac{\partial F_\theta}{\partial u}\right|$ and tangent field $T_\theta(z)$ for the saddle-node/pitchfork models for some set values of $\theta$ revealing that $\left|\frac{\partial F_\theta}{\partial u}\right| = 0$ defines bifurcations

Figure A.2 reveals that $\left|\frac{\partial F_\theta}{\partial u}\right| = 0$ is also a level set and that the intersection with level set $F_\theta(z) = 0$ defines the bifurcations at specific parameter $\theta$. In this particular setting we can see that the tangent field $T_\theta(z)$ only folds when $\left|\frac{\partial F_\theta}{\partial u}\right| = 0$. Plotting the value of the determinant along $F_\theta(z) = 0$ from Figure A.2 would give rise to Figures 1. The directional derivative of the determinant $\left|\frac{\partial F_\theta}{\partial u}\right|$ along the tangent field $T_\theta(z)$ is defined as

$$\frac{d}{ds}\left|\frac{\partial F_\theta}{\partial u}\right| := \hat{T}_\theta(z) \cdot \frac{\partial}{\partial z}\left|\frac{\partial F_\theta}{\partial u}\right| \tag{A.7}$$

where $\hat{T}_\theta(z)$ is the unit tangent field.

## B  Bifurcation Measure Properties

Consider a vector $v(s) \in \mathbb{R}^N$ parametrised by $s \in \mathbb{R}$ that is tangent to an equilibrium manifold defined by $F_\theta(u) = 0$. The conditions for a non-degenerate static bifurcation at $s^*$ along such a tangent can be expressed in terms of an eigenvalue $\lambda(s)$ of the state-space Jacobian crossing zero with a finite slope. A bifurcation exists at $s^*$ if

$$\frac{\partial F_\theta}{\partial u}v(s) = \lambda(s)\,v(s) \quad \exists \lambda: \quad \lambda(s)|_{s=s^*} = 0 \qquad \left.\frac{d\lambda}{ds}\right|_{s=s^*} \neq 0 \tag{B.1}$$

These conditions are necessary and sufficient for a non-degenerate static local breakdown of stability. For now we do not consider dynamic bifurcations involving limit cycles or imaginary parts of eigenvalues and restrict $\lambda \in \mathbb{R}$. Cases where both $\lambda(s)|_{s=s^*} = 0$ and $\left.\frac{d\lambda}{ds}\right|_{s=s^*} = 0$ require investigation into higher order derivatives $\frac{d^n \lambda}{ds^n}$. These are the cases we refer to as *degenerate* and are not considered here.

Instead of considering conditions on each eigenvalue individually it is possible to use the determinant of the state-space Jacobian to detect whether the conditions (B.1) are satisfied. The determinant can be expressed as the product of eigenvalues

$$\left|\frac{\partial F_\theta}{\partial u}\right| = \prod_{n=1}^{N} \lambda_n(s) \tag{B.2}$$

Applying the product rule when differentiating yields

$$\frac{d}{ds}\left|\frac{\partial F_\theta}{\partial u}\right| = \sum_{n=1}^{N} \frac{d\lambda_n}{ds} \prod_{n'\neq n} \lambda_{n'}(s) \tag{B.3}$$

$$= \left|\frac{\partial F_\theta}{\partial u}\right| \sum_{n=1}^{N} \frac{d\lambda_n}{ds} \lambda_n(s)^{-1} \tag{B.4}$$

Substituting this expression into measure (5)

$$\varphi_\theta(s) = \left(1 + \left|\sum_{n=1}^{N} \frac{d\lambda_n}{ds} \lambda_n(s)^{-1}\right|^{-1}\right)^{-1} \tag{B.5}$$

Which implies the following

$$\exists \lambda : \quad \begin{cases} \lambda(s) = 0 & \frac{d\lambda}{ds} \neq 0 \\ \lambda(s) \neq 0 & \frac{d\lambda}{ds} \to \pm\infty \end{cases} \implies \varphi_\theta(s) = 1 \tag{B.6}$$

If there exists an eigenvalue that satisfies conditions (B.1) then the measure is equal to one. The measure also approaches one in cases where the rate of change of an eigenvalue with respect to a manifold $s$ location diverges while not crossing zero. This gives rise to finite gradients in the eigenvalue term in regimes far away from any bifurcation.

## C   Leibniz Rule for Space Curves

Suppose there exists a one dimensional space curve $\mathcal{C}(\theta)$ embedded in $z \in \mathbb{R}^{N+1}$ whose geometry changes depending on input parameters $\theta \in \mathbb{R}^M$. This curve could be open or closed and changes in $\theta$ could change the curve topology as well. Let the function $\gamma_\theta : \mathbb{R} \to \mathbb{R}^{N+1}$ be a parametrisation of the position vector along the curve within a fixed domain $s \in \mathcal{S}$. Note that the choice of parametrisation is arbitrary and our results should not depend on this choice. Furthermore, if we parametrise the curve $\mathcal{C}(\theta)$ with respect to a fixed domain $\mathcal{S}$ the dependence on $\theta$ is picked up by the parametrisation $\gamma_\theta(s)$. We can write a line integral of any scalar function $L_\theta : \mathbb{R}^{N+1} \to \mathbb{R}$ on the curve as

$$L(\theta) := \int_{\mathcal{C}(\theta)} L_\theta(z) \, \mathrm{d}z = \int_{\mathcal{S}} L_\theta(z) \left|\frac{d\gamma_\theta}{ds}\right| \mathrm{d}s \; _{z=\gamma_\theta(s)} \tag{C.1}$$

where $\left|\frac{d\gamma_\theta}{ds}\right|$ is the magnitude of tangent vectors to the space curve and we remind ourselves that the integrand is evaluated at $z = \gamma_\theta(s)$. We would like to track how this integral changes with respect to $\theta$. The total derivative with respect to $\theta$ can be propagated into the integrand [34] as long as we keep track of implicit dependencies

$$\frac{dL}{d\theta} = \int_{\mathcal{S}} \left|\frac{d\gamma_\theta}{ds}\right| \left(\frac{\partial L}{\partial \theta} + \frac{\partial L}{\partial z} \cdot \frac{dz}{d\theta}\right) + L_\theta(z) \frac{d}{d\theta}\left|\frac{d\gamma_\theta}{ds}\right| \mathrm{d}s \; _{z=\gamma_\theta(s)} \tag{C.2}$$

Here we applied the total derivative rule in the first term due to the implicit dependence of $z$ on $\theta$ through $z = \gamma_\theta(s)$. Applying the chain rule to the second term

$$\frac{d}{d\theta}\left|\frac{d\gamma_\theta}{ds}\right| = \left|\frac{d\gamma_\theta}{ds}\right|^{-1} \frac{d\gamma_\theta}{ds} \cdot \frac{d}{d\theta}\left(\frac{d\gamma_\theta}{ds}\right) \tag{C.3}$$

By choosing an $s$ that has no implicit $\theta$ dependence we can commute derivatives

$$\frac{d}{d\theta}\left(\frac{d\gamma_\theta}{ds}\right) = \frac{d}{ds}\left(\frac{d\gamma_\theta}{d\theta}\right) \quad \Rightarrow \quad \frac{d}{d\theta}\left|\frac{d\gamma_\theta}{ds}\right| = \left|\frac{d\gamma_\theta}{ds}\right|^{-1} \frac{d\gamma_\theta}{ds} \cdot \frac{d}{ds}\left(\frac{d\gamma_\theta}{d\theta}\right) \tag{C.4}$$

To proceed we note that the unit tangent vector can be written as an evaluation of a tangent field $\hat{T}_\theta(z)$ defined in the whole domain $z \in \mathbb{R}^{N+1}$ along the parametric curve $z = \gamma_\theta(s)$. The unit tangent field may disagree with the tangent given by $\frac{d\gamma_\theta}{ds}$ up to a sign

$$\hat{T}_\theta(z)\Big|_{z=\gamma_\theta(s)} = \pm\left|\frac{d\gamma_\theta}{ds}\right|^{-1} \frac{d\gamma_\theta}{ds} \tag{C.5}$$

this leads to

$$\frac{d}{d\theta}\left|\frac{d\gamma_\theta}{ds}\right| = \left|\frac{d\gamma_\theta}{ds}\right|\left(\hat{T}_\theta(z)\cdot\frac{\partial}{\partial z}\left(\frac{d\Gamma_\theta}{d\theta}\right)\cdot\hat{T}_\theta(z)\right)_{z=\gamma_\theta(s)} \tag{C.6}$$

It is possible to find the normal deformation of the implicit space curves due to changes in $\theta$. This can be done by taking the total derivative of the implicit equation defining the level set

$$\frac{dF_\theta(z)}{d\theta} = \frac{\partial F}{\partial\theta} + \frac{\partial F}{\partial z}\cdot\frac{dz}{d\theta} \tag{C.7}$$

We can rearrange for $\frac{dz}{d\theta}$ using the Moore-Penrose inverse of the rectangular Jacobian matrix $\frac{\partial F}{\partial z}$ which appeared in equation (A.2). Since the level set is defined by $F_\theta(z) = 0$ the total derivative along the level set $dF_\theta(z) = 0$ and we arrive at an expression for the deformation field [28]

$$\frac{dz}{d\theta} = -\frac{\partial F}{\partial z}^\top\left(\frac{\partial F}{\partial z}\frac{\partial F}{\partial z}^\top\right)^{-1}\frac{\partial F}{\partial\theta} \tag{C.8}$$

The tangential component of the deformation field is not uniquely determined because there is no unique way of parametrising a surface. This is the subject of many computer graphics papers [28, 36, 37]. We are however not interested in the continuous propagation of a mesh - as is the subject of those papers. In fact we are looking for a deformation field that is orthogonal to the tangent vector $\hat{T}_\theta(z)\cdot\frac{dz}{d\theta} = 0$ for the space curve, and therefore letting the tangential component of the deformation equal zero is a valid choice and we can it instead of the parametrised deformation

$$\frac{d\gamma_\theta}{d\theta}\to\frac{dz}{d\theta} \tag{C.9}$$

To summarise we now have the gradient of our line integral only in terms of the implicit function defining the integration region.

$$\frac{dL}{d\theta} = \int_{F_\theta(z)=0}\frac{\partial L}{\partial\theta} + \frac{\partial L}{\partial z}\cdot\varphi_\theta(z) + L_\theta(z)\,\hat{T}_\theta(z)\cdot\frac{\partial\varphi}{\partial z}\cdot\hat{T}_\theta(z)\,\mathrm{d}z \tag{C.10}$$

$$\text{where}\quad \hat{T}_\theta(z) := \frac{T_\theta(z)}{|T_\theta(z)|}\qquad T_\theta(z) := \left|\begin{matrix}\hat{z}\\\partial_z F_\theta\end{matrix}\right|\qquad \varphi_\theta(z) := -\frac{\partial F}{\partial z}^\top\left(\frac{\partial F}{\partial z}\frac{\partial F}{\partial z}^\top\right)^{-1}\frac{\partial F}{\partial\theta} \tag{C.11}$$

We have settled on choosing normal deformations which we will call $\varphi_\theta(z)$. The above result can be seen a the generalised Leibniz rule [34] for the case of line integration regions. The last integrand term can be seen as the divergence the vector field $\varphi_\theta(z)$ projected onto the one dimensional space curve.

## D   Application of Bifurcation Inference to a Complex Model

To demonstrate the wider reaching applicability of our method we optimise the *double exclusive reporter* [3], a synthetic gene circuit in *E. coli* that was designed to exhibit a cusp bifurcation. The circuit behaviour is observed by measuring a fluorescent protein whose expression is controlled by transcription factors (regulatory proteins) LacI ($L$) and TetR ($T$), whose expression is in turn controlled by externally controllable *input* signals $c_6$ and $c_{12}$. To apply the method, we consider one of the input signals be the control condition $c_6 = p$, with the other packed together with the remaining 20 parameters into vector $\theta$. Once the optima $\theta^*$ have been obtained, we perform dimensionality reduction using `GigaSOM.jl` [38] so that the results can be visualised in a two dimensional embedding (Figure D.1A).

The embedding reveals four optimal parameter regions. We find that, as with the two-state model in the main text (11), there are two qualitatively distinct regimes: mutual activation (region 1) and inhibition (regions 2-4). The mutual inhibition region can be further subdivided into three regions that are geometrically equivalent, but kinetically distinct: region 3 has swapped kinetic roles for regulatory proteins LacI and TetR compared to region 2, and region 4 has additional damped oscillations in the dynamics across the whole range of *input* $c_6$ (Figure D.1B). The two dimensional embedding of sampled optima $\theta^*$ enables navigation the space of qualitative behaviours of the *double exclusive reporter* and organisation in terms of geometric and kinetic equivalence.

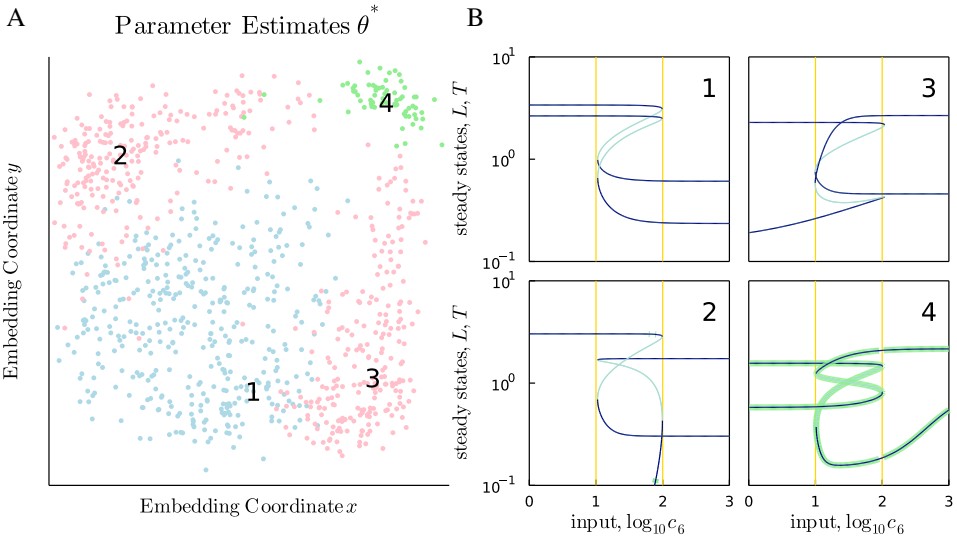

Figure D.1: Bifurcation inference for the *double exclusive reporter*. A. Optimal parameter estimates $\theta^*$ for the targets $\mathcal{D} = \{1, 2\}$ (indicated by yellow lines in panel B) reveal four regions with two geometrically different regimes: mutual activation (region 1) and mutual inhibition (regions 2-4). B. Example bifurcation diagrams indicate that region 2 has swapped kinetics between $L$ and $T$ to region 3. Region 4 has models with non-zero imaginary parts to eigenvalues indicating damped oscillations (shown in light green).

These results were obtained with a modification of the bifurcation measure (5) to improve convergence rates. In parameter regimes where bifurcations are not present, according to conditions (B.6), maximising the measure $\varphi_\theta(s)$ can lead to a divergence in directional derivative $\frac{d\lambda}{ds} \to \pm\infty$ rather than a creation of a bifurcation. To discourage this from happening we can flatten out the gradients in that regime by applying the `tanh` non-linearity to the determinant. This leads to

$$\varphi_\theta(s) := \left(1 + \left|\frac{\tanh\left|\frac{\partial F_\theta}{\partial u}\right|}{\frac{d}{ds}\tanh\left|\frac{\partial F_\theta}{\partial u}\right|}\right|\right)^{-1} \tag{D.1}$$

# E   Extension for Hopf Bifurcations

In order to detect bifurcations involving limit cycles, the measure must be extended to detect changes in the real part $\Re e[\lambda(s)]$ for any eigenvalue of the Jacobian. These conditions can no longer be compactly written in terms of the determinant. Instead, the measure can be defined as the sum of eigenvalue terms

$$\varphi_\theta(s) := \sum_{\lambda(s) \in \frac{\partial F_\theta}{\partial u}} \left( \left| \frac{d}{ds} \log \Re e[\lambda(s)] \right|^{-1} + 1 \right)^{-1} \tag{E.1}$$

The directional derivative of the logarithm diverges under two conditions: when eigenvalues vanish $\lambda(s) = 0$ and when the directional derivative $\frac{d}{ds}\Re e[\lambda(s)]$ diverges. These properties are sufficient for detecting the onset of damped oscillations and emergence of limit cycles via Hopf bifurcation as shown in Figure E.1. Eigenvalues with negative real part which gain a finite imaginary part give rise to damped oscillations. At this onset we observe a discontinuity in the derivative $\frac{d}{ds}\Re e[\lambda(s)]$ which is detected by equation (E.1). Once damped oscillations exist, flipping the stability of the stable fixed point gives rise to a limit cycle, which can be detected by inspecting $\Re e[\lambda(s)]$.

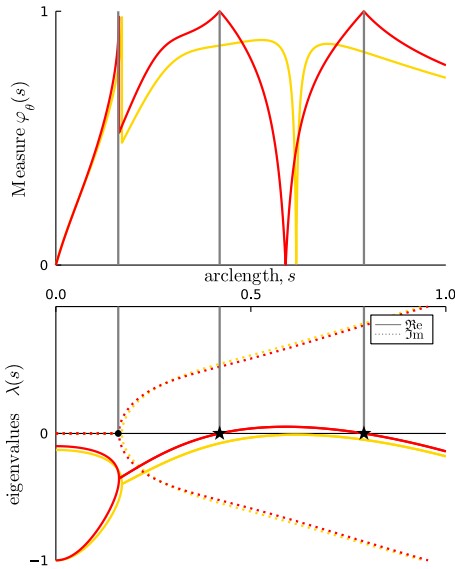

Figure E.1: Bifurcation measure $\varphi_\theta(s)$ and eigenvalues $\lambda(s)$ along the arclength $s$ for two different bifurcation curves demonstrating how the measure detects non-zero imaginary parts $\Im m[\lambda]$ (onset of damped oscillations marked by circle) and sign changes in real parts $\Re e[\lambda]$ (Hopf bifurcations marked by stars)

In principle it is possible to construct measures to detect a variety of bifurcations as long as the conditions can be expressed in terms of derivatives with respect to fixed-point manifold direction $s$. Measures can be used sequentially or in parallel to encourage optimisers to run through a sequence of bifurcations or place specific bifurcation types next to each other.