# OpenReview forum: "Parameter Inference with Bifurcation Diagrams"
_NeurIPS.cc/2021/Conference — NeurIPS 2021 Poster_

### Official Review · Reviewer_L17k · 2021-06-24

**Rating:** 8
**Confidence:** 4

**Summary:**

The paper introduces a method for fitting parameterised differential equations to qualitative behaviour: the location and number of bifurcations. Several new techniques are introduced: a loss for matching bifurcations to target locations, a loss for producing the correct number of bifurcations, and finally an approach to differentiating these terms via implicit differentiation methods.

**Limitations And Societal Impact:**

No discussion is included, but I think this is fine for a work of this type.

**Main Review:**

### Summary

This paper is excellent.

The problem considered is difficult, interesting, and has a clear audience.

The technical contributions of the paper include several techniques, which to the best of my knowledge are new, technically sound, and frankly quite ingenious.

The clarity of the presentation is generally good. I was able to understand the paper without too much difficulty. I do have some suggestions for improving this below.

For these reasons I definitely recommend acceptance.

### Discussion

The experiments are undoubtedly the weakest part of the paper. Only two are considered (three including the complexity study), and these are on relatively simple synthetic problems. Extensions to less toy, practical, problems would substantially strengthen the paper. In other words, what are some impressive use cases that help sell this technique?

The differential equations considered are essentially straightforward traditional parameterised differential equations. Do the authors envisage applications to neural differential equations, no doubt on more complicated problems? Perhaps combined with a traditional supervised loss to fit the quantitative behaviour. For example, much of my own work involves fitting neural differential equations to complicated dynamical systems, for which no mechanistic description is available -- only broad qualitative descriptions. It would certainly pique my interest to see an example of this type.

Given that there is a little extra space at the end of the paper: I think some (brief, intuitive) discussion on the steady-state solver(s) would be the most helpful piece of additional background material to include.

### Minor points

Line 86, equation (5), and elsewhere: $\frac{d}{ds}|\frac{\partial F_\theta}{\partial u}|$ does not feature an $s$ to differentiate with respect to. Moreover the definition of the curve parameterised by $s$ (presumably a level set of $F_\theta$) is not made clear. Leaving these details implicit hurts readability.

Equation (2) is lacking a dividing line between the $p$ and the $\exists$.

The upper graphs in Figure 1 could do with a y-axis label $u$.

Section 3.1: it would be helpful to explicitly reference the model (from earlier) which is being optimised. The optimiser used is not stated.

I do not think I would describe the approach as semi-supervised, or that $(|P| - |D|)\log\Psi$ represents an unsupervised loss. Whilst these terms are often blurred and without precise meaning: I would take semi-supervised learning to mean that some of the training data could have had a label, but is missing. (For example, training to perform MNIST image classification, except only some 10% of the data has its label.) Likewise I would describe $(|P| - |D|)\log\Psi$  as a supervised loss: it pushes together $|P|$ and $|D|$.

**Time Spent Reviewing:**

4

---

> ### Author Response · Authors · 2021-08-10
> **Neural ODEs example**
>
> The authors are delighted at the reviewer's enthusiastic response and look forward to discussions on neural differential equations ahead. We propose running the method on a simple neural ODE
> $$
> \frac{du}{dt} = h(\mathbf{W}u) + \mathbb{1}_k \\,p
> $$
> where $u\in\mathbb{R}^N$ is a vector $\mathbf{W}$ is an $N\times N$ matrix containing all the parameters to be learned, $h$ is an element-wise activation and $\mathbb{1}_k$ is a one-hot vector with a non-zero element at the $k$th position. Depending on how much space there is and how much time we are given to respond, this may be included in an appendix. Also, as mentioned in the response above, we will attempt to run the method on a more complicated published model (12 states, 40 parameters), to further test the generality of our approach.
> ## Steady State Solvers
> We thank the reviewer for the suggestion to include some intuitive description of the steady-state solver and parameter continuation methods, to help improve the accessibility of the article. We will happily provide a brief description in a revised manuscript. More details on parameter continuation methods, and specifically deflated continuation will be provided in an appendix.
> ## Minor points
> We agree with all minor points the reviewer has made. We propose new names to the two terms in the cost function:
> - Error term (or goodness of fit term)
> - Eigenvalue term (as it is driven by the eigenvalues of the jacobian)

---

> > ### Comment · Reviewer_L17k · 2021-08-15
> > **Response**
> >
> > Thank you for your response.
> >
> > I think the proposed neural ODE may be slightly too simple a model for many problems. I would probably suggest either
> > $$
> > \frac{\mathrm{d} u}{\mathrm{d} t}(t) = W_1 h(W_2 u + b_2) + b_1 + \mathbb{1}_k p
> > $$
> > i.e. an MLP as the neural component, or
> > $$
> > \frac{\mathrm{d} u}{\mathrm{d} t}(t) = \tanh(W_1 h(W_2 u + b_2) + b_1) + \mathbb{1}_k p
> > $$
> > which on some problems can be helpful to avoid solutions changing too quickly -- e.g. this can be used to manage untrainably large initial losses.
> >
> > Here $W_1 \in \mathbb{R}^{M \times N}$, $b_1 \in \mathbb{R}^N$, $W_2 \in \mathbb{R}^{N \times M}$, $b_2 \in \mathbb{R}^M$. For many physical problems them $N, M$ can be very small, e.g. $N=2$, $M=8$.
> >
> > In any case, based on the authors' response -- both to me and Reviewer DHJo -- I am happy to maintain my score. Nice work!

---

### Official Review · Reviewer_cqjt · 2021-07-16

**Rating:** 6
**Confidence:** 3

**Summary:**

The authors present an approach to inference of parameters of a limited scope of dynamical systems with codimension one bifurcations from user-defined number and location of bifurcation points. The approach solves the parameter inference problem by gradient-based optimization function consisting of two terms. A term matching the provided to the estimated bifurcation locations and a term forcing desired properties of the determinant of the Jacobian of the system. The approach is demonstrated on two simple examples, minimal models and a synthetic toggle switch.

**Ethical Concerns:**

There are no ethical issues with this paper.

**Limitations And Societal Impact:**

The presented work doesn't have negative societal impact.

The authors acknowledge the limitations of their work adequately. The main limitation that is not addressed however is the limited applicability of the approach. See main review.

**Main Review:**

The paper is well organized and written in a clear and understandable manner. The task that the authors aim to solve is, in general, relevant. Their solution somewhat novel. Previous work on solving this task exists and the authors acknowledge it but don't consider it for direct comparison. Previous work on biasing a goodness of fit with domain knowledge drive the optimization exists, especially coming from the domain of modeling dynamical systems for systems and synthetic biology, however it is not reported in this paper.

The definition of the task being solved as being semi-supervised might not be appropriate, as it doesn't fit the frame defined by the machine learning community, from where the term is borrowed. The inference of parameters of dynamical systems, given measurements, qualitative or quantitative descriptions of desired or expected behavior better fits a supervised learning task. The task definition and the theoretical analysis are sufficient and convincing.

The aimed application of the presented work is to inverse problems from biology or engineering. However, the presented applications are toy models and a small synthetic biology model. Inverse problems in biology in general are much more complex than the examples presented in this work and rarely fall in the scope of dynamical systems with codimension one bifurcation. Furthermore, inverse problems from the domain of biology have consistently been proven to be difficult to solve due to being underdetermined, lack of data or various sources of noise. Although the authors acknowledge the model selection problem, they address it superficially in the conclusion section. Additionally, although their minimal models are extensible in the number of parameters they do not analyse their impact on the computational cost. The application of the presented approach to such real world problems is therefore severely limited.


**Time Spent Reviewing:**

3

---

> ### Author Response · Authors · 2021-08-10
> **Adding direct comparisons and more complex model**
>
> We would like to thank you for your helpful review!
>
> ### Direct comparisons
> To the best of our knowledge this is the first end-to-end gradient based method for finding and matching bifurcations and therefore a direct comparison with previous works is difficult. However, we propose the following baseline experiments to elucidate the benefits of the fundamental aspects of our approach:
> - Identical cost function but optimised with gradient-free optimiser: Nelder Mead. This would shows us the speed-up with respect to non-gradient based approaches.
> - Replacing the unsupervised term with a random sampling method. Such a method is implemented in Oscill8, and could be used to quantify the value of encouraging the optimiser to move to regions of parameter space that contain bifurcations. However, it is important to note that such a comparison would be somewhat subjective, as the value of the unsupervised term would increase as the fraction of the parameter space that contains bifurcations also increases.
>
> We could add these two comparisons to our computational complexity study, where we have shown that our method scales like $N^2$ with respect the the number of states (Figure 5).
>
> ### Missing literature references
> We acknowledge there are publications that bias goodness of fit with domain knowledge. We are happy to include 3 references to flesh out the "similar works" paragraph of the introduction. We would like to verify that we correctly understood the vein of literature the reviewer is referring to, but would like to check with the chairs if it is acceptable and preserves anonymity to link the reviewer to the 3 examples (not published by us).
>
> ### More complex model example
> We propose running our method on a previously published ODE model that was used to design a quorum sensing network in bacteria, and is significantly more complex than the models analysed in the original submission (12 states, 40 parameters). We do not include a reference to this model to preserve anonymity.
>
> We also propose to include analysis of the method applied to a neural ODE example (see reply to reviewer L17k), the complexity of which could be controlled by increasing the dimensionality of the state vector, similar to how we tested the method on models of increasing size in Figure 5.
>
> Please do offer guidance on whether either or both of these would improve the presentation of our work.
>
> ### Renaming terms in cost function
> We agree that describing the method as semi-supervised is not appropriate and would like to propose new names to the two terms in the cost function:
> - Error term (or goodness of fit term)
> - Eigenvalue term (as it is driven by the eigenvalues of the jacobian)
>
> ### Broader Impact
> We acknowledge the challenge of under-determined inverse problems will prevail with this method. This method can be extended to co-dimension two bifurcations by using a marching cubes algorithm to solve for the steady state surface; equations for gradients (10) become under-determined, but the normal component of derivatives can still be calculated (references on line 276) and suffice for optimisation.
>
> We can also include a more detailed discussion on how to extend measure (5) for other bifurcation types such as Hopf. While this article on its own may not demonstrate wide reaching applicability, it lays the foundations for a powerful approach that does have far reaching applicability

---

> > ### Comment · Reviewer_cqjt · 2021-08-30
> > **Response**
> >
> > I believe that what the authors are suggesting will improve the presentation of their work and that the paper can be accepted. However, as more details of their suggested changes are not available to me for anonymity preserving reasons, I leave the decision at the discretion of the Chairs.

---

### Official Review · Reviewer_DHJo · 2021-07-26

**Rating:** 7
**Confidence:** 4

**Summary:**

In this work, authors suggest a gradient-based semi-supervised approach to infer the parameters of ODEs for producing a bifurcation diagram. A bifurcation measure is defined that uses the determinant of the state-space Jacobian as an indicator for bifurcating parameter regimes in the unsupervised term of the cost function. They demonstrate parameter inference with minimal models which explore the space of saddle-node and pitchfork diagrams and the genetic toggle switch from synthetic biology.

**Limitations And Societal Impact:**

Yes

**Main Review:**

1. The results of this work seem very interesting and important
2. The paper is well written and clear; but some math parts still need more clarification
3. I should point out some comments that need to be addressed (please see below):

- In line 79, authors have been mentioned that "a co-dimension
one bifurcation can be defined by a set of conditions on the determinant of the Jacobian |∂F_θ\∂u|". But, based on Sotomayor's Theorem (for codimension-1 local bifurcations), it is only one sufficient condition and to investigate such bifurcations we also need some conditions on "∂^2F_θ /∂u^2" and "∂F_θ\∂p" (although they have discussed such conditions briefly in Appendix). So, this sentence is misleading & confusing.

- In Appendix B, it is mentioned "If we are not specifically looking for transcritical or pitchfork bifurcations, it is sufficient to consider a non-zero Hessian determinant and non-zero vector slope (Eq. (21))".  But, it is not correct; please see Sotomayor's Theorem for more clarity. Especially, since for saddle-node bifurcations the conditions in Eq.(21) are not enough and such conditions depend on **left eigenvectors** as well.

- Although the authors have mentioned different kinds of bifurcations, from Eq.(2) the type of bifurcation is not clear and they have only discussed relationships between conditions (2) and the conditions for saddle-node bifurcation. Is it possible to add any other conditions from which the type of bifurcation is determined as well? Otherwise, the authors should clarify that (2) is related to only one bifurcation e.g. saddle-node bifurcation.

- In line 91, it is mentioned "Pitchfork bifurcations are special cases of the saddle-node where a single steady state splits into two stable and one unstable steady state (Figure 1B)." But, this is mathematically **incorrect** since pitchfork bifurcations are **not** special cases of the saddle-node bifurcations; please see Sotomayor's Theorem for more clarity.

**Time Spent Reviewing:**

7 hours

---

> ### Author Response · Authors · 2021-08-07
> **revised Appendix B for mathematical clarification**
>
> Thank you for the thorough read of our paper! We agree that mathematical clarifications are needed in connecting our conditions (2) with well-known definitions of the bifurcations under study. We would also like to thank the reviewer for pointing us towards Sotomayor's Theorem for a vectorised version of the well-known conditions for static bifurcations.
>
> To address the reviewer's first two points we would like to propose the following new version of Appendix B which focuses on the how the bifurcation measure can detect breakdowns of structural stability a.k.a. zero crossings in Jacobian eigenvalues. This would replace any discussion on conditions for saddle-nodes vs pitchforks or trans-critical.
>
> ## Bifurcation measure properties
>
> Consider a vector $v(s)\in\mathbb{R}^N$ parametrised by $s\in\mathbb{R}$ that is tangent to an equilibrium manifold defined by $F_{\theta}(u)=0$. The conditions for a non-degenerate static bifurcation at $s^*$ along such a tangent can be expressed in terms of an eigenvalue $\lambda(s)$ of the state-space Jacobian crossing zero with a finite slope. A bifurcation at $s^*$ exists at if
> $$
>     \frac{\partial F_{\theta}}{\partial u}v(s)=\lambda(s) \\,v(s)
>     \quad\exists\lambda:\quad
>     \left.\lambda(s)\right|_{s=s^*}=0
>     \qquad
>     \left.\frac{d\lambda}{ds}\right|\neq 0
> $$
> These conditions are necessary and sufficient for a static local breakdown of stability. For now we do not consider limit cycles and or imaginary parts of eigenvalues and restrict $\lambda\in\mathbb{R}$.
>
> Instead of considering conditions on each eigenvalue individually it is possible to use the determinant of the state-space Jacobian to detect whether the conditions are satisfied. The determinant can be expressed as the product of eigenvalues
>
> $$
>     \left| \frac{\partial F_{\theta}}{\partial u} \right|=\prod_{n=1}^N\lambda_n(s)
> $$
> Applying the product rule when differentiating yields
> $$
>     \frac{d}{ds} \left| \frac{\partial F_{\theta}}{\partial u} \right| =
>     \sum_{n=1}^N\frac{d\lambda_n}{ds}\prod_{n'\neq n}\lambda_{n'}(s)
> $$
> $$
>     = \left| \frac{\partial F_{\theta}}{\partial u} \right| \sum_{n=1}^N\frac{d\lambda_n}{ds}\lambda_{n}(s)^{-1}
> $$
> Substituting this expression into measure
> $$
>     \varphi_{\theta}(s)=
>     \left(1+\left|\sum_{n=1}^N\frac{d\lambda_n}{ds}\lambda_{n}(s)^{-1}\right|^{-1}\right)^{-1}
> $$
> Which implies the following limits
> $$
>     \varphi_{\theta}(s)=1 \quad\iff\quad\exists\lambda:\quad
>     \\,\lambda(s)=0 \quad\frac{d\lambda}{ds}\neq 0
> $$
> $$
>     \varphi_{\theta}(s)=1 \quad\iff\quad\exists\lambda:\quad
>     \\,\lambda(s)\neq0 \quad\frac{d\lambda}{ds}\rightarrow\pm\infty
> $$
> The measure is equal to one if there exists an eigenvalue that satisfies condition for the breakdown of stability.
>
> The measure also approaches one in cases where the rate of change of an eigenvalue with respect to a manifold location $s$ diverges while not crossing zero. This gives rise to finite gradients in the unsupervised term in regimes far away from any bifurcation.
>
> ## Other types of bifurcations
> Addressing the reviewer's third point. We have an example of a measure that detects Hopf bifurcations and can include this in discussion. In principle to bias the optimisation towards different types of bifurcations one has to design a measure
> $$
>     \varphi_{\theta}(s)=\frac{1}{1+g_{\theta}(s)}
> $$
> where $g(s)$ is a scalar positive semi-definite function that diverges when all the conditions for the chosen bifurcation type are satisfied.
> ## Perfect pitchforks are rare
> Addressing the reviewer's fourth point: the intent of the authors with the statement on line 91 was to point out that a perfect pitchfork can only be found for $\theta_1=0$ and small perturbations to $\theta_1$ lead to imperfect pitchforks, which are actually saddle-nodes. This statement, however, does not seem to add value to the paragraph so can be removed.

---

> > ### Comment · Reviewer_DHJo · 2021-08-14
> > **Thanks for your response!**
> >
> > Thanks for the clarification! Most of my concerns are addressed and I would like to recommend acceptance. However, I need to discuss some minor points before updating my score. Here are the points:
> >
> >  1) If by "breakdown of stability" you mean change of stability related to bifurcation, then the conditions
> >
> > ∂F_θ/∂u v(s) =λ(s)v(s)     \\   ∃λ:  λ(s)|s=s∗= 0      \\         dλ/ds \neq 0
> >
> > are of course sufficient conditions for change of stability; but how they can be necessary conditions as well? For more clarity, let λ(s) = s^3. Then, obviously, λ(0)=0 and \frac{dλ}{ds}  at  "s=0"  is zero but the sign of  λ changes passing through 0 (which implies the stability changes through s=0).
> >
> > 2) By  \frac{dλ}{ds} \neq 0   I assume you mean the mentioned derivative is non-zero at "s=s^*"  but not everywhere, right?
> >
> > 3) Do you mean two conditions
> >
> > φ_θ(s) = 1  ⇐⇒  ∃λ:  \\  λ(s)= 0  \\  dλ/ds \neq 0
> >
> > and
> >
> > φ_θ(s) = 1  ⇐⇒  ∃λ:  \\  λ(s) \neq  0  \\  dλ/ds →±∞
> >
> > are equivalent? If so, how they can be equivalent?

---

> > > ### Author Response · Authors · 2021-08-14
> > > **"degenerate" breakdown of stability**
> > >
> > > 1. Indeed in cases where there is a change in stability at $s^*$ but both $\lambda=0$ and $\frac{d\lambda}{ds}=0$ at $s^*$, then we would expect one of the higher order derivatives $\frac{d^n\lambda}{ds^n}\neq0$ where $n$ is odd. Let us call these cases "degenerate". Our paper should make it clear that the bifurcation measure $\varphi_{\theta}(s)$ can only detect "non-degenerate" breakdowns of stability
> > > 2. Yes. the formula should have read $\frac{dλ}{ds}\big|_{s=s^*} \neq 0$ apologies for the typo
> > > 3. They are not equivalent. The two conditions should actually have $\Leftarrow$ instead of $\iff$

---

> > > > ### Comment · Reviewer_DHJo · 2021-08-14
> > > > **My concerns are addressed well**
> > > >
> > > > Thanks for your explanations! They address my concerns. I have updated my score to reflect this.

---

### Decision · Program_Chairs · 2021-09-27

**Decision:**

Accept (Poster)

**Comment:**

All reviewers agree that the paper propose an interesting approach to the problem of estimating parameters in differential equations. Although some reviewers have some technical concerns at their first reviews, they satisfy the authors' responses to address those. It seems that, although there are some points that should be modified from the current form including the terms used in the paper (such as semi-supervised learning), I think we can expect the authors modify the paper in the camera-ready by reflecting the discussion. Based on these, I recommend acceptance (poster) for this paper.